# Uncovering the relationship between working memory and performance in the Jigsaw classroom

Eva Vives[1,2*], Marco Bressan[2], Céline Poletti[3], Denis Caroti[4], Fabrizio Butera[3], Pascal Huguet[5], ProFAN consortium[¶], Isabelle Régner[2]

1 Moral & Social Brain Lab, Department of Experimental Psychology, Ghent University, Belgium, 2 Aix Marseille Univ, CNRS, CRPN, Marseille, France, 3 Institut de Psychologie, Université de Lausanne, Lausanne, Suisse, 4 Aix Marseille Univ, CNRS, Centre Gilles-Gaston Granger, Aix-en-Provence, France, 5 Université Clermont Auvergne, CNRS, LAPSCO, F-63000 Clermont-Ferrand, France

¶ ProFAN consortium: the complete membership of the author group can be found in the Acknowledgments section.
* eva.vives@ugent.be

## Abstract

The Jigsaw classroom is a popular cooperative learning method based on resource interdependence, which requires students to work in small groups on complementary pieces of information, to achieve learning. However, Jigsaw classroom is characterized by contradictory findings and a lack of knowledge on its underlying cognitive mechanisms.

The present study examined whether working memory capacity, a key executive function for academic achievement and learning, mediated or moderated the effects of Jigsaw classroom on individual performance. Undergraduate students ($n == 342$) attending French University took part in this study ($Mage == 19.40$, $SD == 1.21$, 60% female). Students worked in small groups on a critical thinking reasoning task, either in the Jigsaw condition or in a cooperative (control) condition without resource interdependence. Working memory was assessed twice, before and during the group activities, by using a complex working memory span task. We analyzed students' individual score to a quiz on logical fallacies. Multilevel analyses revealed that working memory capacity moderated—but did not mediate—the effect of the Jigsaw classroom. That is, Jigsaw enhanced performance for students with low working memory capacities. These findings offer insight into the potential cognitive mechanisms implied in the success of the Jigsaw method and provide new recommendations for educators on how to redeem the deficit of low working-memory-capacity students on performance.

## Introduction

Research on cooperative learning gained momentum in the early 1980s [1], demonstrating positive effects on academic achievement [2–4]. Among the various methods developed to implement small-group learning in schools, the Jigsaw classroom, designed by Aronson et al. [5], stands out as one of the most popular [6]. Originally developed to reduce racial conflict

**Data availability statement:** The data that support the findings of this study are openly available in OSF from the Working Memory and Jigsaw classroom repository at https://osf.io/2vcqf/.

**Funding:** This work was supported by the French Ministry of National Education, Youth and Sports (MENJS); the Ministry of Higher Education, Research and Innovation (MESRI); the « Mission Monteil pour le numérique éducatif », and the « Programme d'investissements d'avenir, expérimentation ProFAN» (PIA). The funders had no role in study design, data collection and analysis, decision to publish, or preparation of the manuscript.

**Competing interests:** The authors have declared that no competing interests exist.

and social inequalities in desegregated US schools, this non-competitive, highly structured peer-learning method promotes positive interdependence and individual accountability among students.

The Jigsaw classroom involves dividing lesson content into smaller pieces, similar to a jigsaw puzzle, and structuring student interactions in small groups through a four-step process. First, each student is assigned to a jigsaw group (also referred to as home group) and receives a unique piece of information about the lesson to read individually. Second, students with the same piece of information form expert groups to understand and master the content. Third, all students return to their original jigsaw groups to teach each other, ensuring that all pieces of the puzzle are gathered, and each student learns the entire lesson. Finally, an individual quiz can be administered to evaluate their learning. Positive resource interdependence and expert groups constitute the cornerstones of the Jigsaw classroom, creating structured and interdependent work environment. Resource interdependence encourages students to actively participate in learning and take responsibility for other's learning. Additionally, the peer-learning phase within expert groups is expected to help low achievers in mastering their assigned content while stimulating high achievers, and strengthening students' responsibility for their group members' learning, fostering mutual interdependence for comprehensive learning.

According to Aronson and Patnoe [7], the Jigsaw classroom was found to enhance students' self-esteem, empathy, and motivation, and to increase academic performance, particularly among disadvantaged students. However, and despite its popularity, the Jigsaw method yields a mix of positive, null, and negative effects on both academic achievement (e.g., [6,8,9]) and psychosocial variables such as self-regulation and self-perceptions (e.g., [10,11]) (for a comprehensive review see Vives and colleagues [12]). Authors also outline that Jigsaw research suffers from methodological shortcomings [6,9,12–14], with findings observed on underpowered studies (for a discussion about potential publication bias, see Stanczak and colleagues [15]).

Beyond these issues, we suggest that taking into account potential moderating and mediating cognitive variables would help to understand Jigsaw mixed findings by specifying for *whom* and *how* the Jigsaw method can benefit (or not) to academic achievement. Given the Jigsaw method's reliance on resource interdependence, investigating working memory (WM) becomes pertinent as this executive function aids complex task performance and new concept learning. Assessing students' WM capacity both before and during the learning process, this study aims to elucidate the role of working memory in moderating and/or mediating Jigsaw's interdependent learning outcomes.

## Inside the black box of Jigsaw classroom

Little research has examined how the Jigsaw method contributes to academic achievement through mediating variables or the specific conditions that enhance its benefits (moderating variables). Studies have shown that group performance [16] and self-efficacy [17] mediate the Jigsaw method's effect on academic achievement. Specifically, Jigsaw's positive impact on individual performance occurs indirectly, through increased group task performance [16] or heightened self-efficacy [17]. Regarding moderating variables, one study demonstrated that students' prior achievement plays a crucial role: the Jigsaw classroom benefited low and medium achievers but not high achievers [18].

Research on collaborative methods, as highlighted by Kirschner and colleagues [19], reveals that task characteristics (e.g., complexity), learner attributes (e.g., collaborative skills), and team factors (e.g., group size) interact with individuals' cognitive load during collaborative tasks. However, cognitive functions have been underexplored in group learning, resulting

in a "black box" impression regarding the impact of collaboration on achievement [20]. This gap is particularly noteworthy in the context of Jigsaw research. Given the necessity of process-oriented research for comprehending the impact of collaboration on individual learning, Kirschner and colleagues advocated for investigating cognitive processes involved in learning activities [19,21–23]). They proposed that sharing information processing within a group could help manage the cognitive load imposed by complex tasks, as task demands often exceed the limited capacity of WM. In such situations, Janssen and colleagues [20] suggested that distributing information processing among group members could alleviate cognitive load on individual WM, requiring fewer WM resources from each student to accomplish the task. Conversely, collaboration on low-demand tasks or among students with sufficient expertise could be at best ineffective and at worst counterproductive.

Hypotheses regarding the role of WM were indirectly supported in experiments where participants reported their perceived cognitive load during tasks [22,24]. Group learners reported higher performance and lower perceived cognitive load compared to individual learners. Only Nebel and colleagues [16] tested whether the Jigsaw method reduced perceived cognitive load through task division among group members. Unexpectedly, students benefited from Jigsaw while reporting higher mental effort compared to control condition (voluntary cooperation, without resource interdependence).

However, little or no work to date has explored the relationships between individuals' cognitive abilities and cooperative learning in the classroom. Given that working memory capacity is a well-established predictor of academic performance and learning capacity during a course, it was critical to explore this "third variable" in the context of the Jigsaw classroom.

## Working memory capacity

The Jigsaw Classroom's material division and peer-learning phase are crucial features for examining WM as a third variable. WM involves storing, manipulating, and retrieving information while minimizing interference within a limited timeframe [25]. This cognitive function is critical for learning and complex cognitive tasks due to its controlled-attention component [26]. WM is positively correlated to academic achievement across domains [27,28] and can be trained over time [29–31]. WM is also a limited-capacity executive process that is sensitive to variations in processing demands [32]. For example, negative stereotypes [33], high-pressure situations [34], evaluative audience [35,36]) can disrupt WM, leading to impaired performance (highlighting WM's mediating role). Other findings have shown that individual differences in WM capacity can moderate the effects of socio-evaluative threats [37–39].

## Present study

As Janssen and colleagues [20] outlined, learning tasks can impose heavy demands on WM, affecting low WM students with attentional challenges [40]. Alternative learning methods are needed to support such students. Recently, Martin and colleagues [41,42] proposed the implementation of load reduction instruction (LRI) in the classroom, incorporating five key principles: difficulty reduction, support and scaffolding, practice, feedback, and guided independence (see Martin and colleagues [43] for an application of LRI in mathematics and English classrooms). Aligned with this significant proposal, we think the Jigsaw Classroom offers a promising approach to alleviate the cognitive burden imposed on students during learning. To address a critical gap in Jigsaw Classroom research, we aim to investigate whether this method can mitigate cognitive load on individual WM and assist students with lower WM resources. This requires assessing WM and testing its mediating and moderating role in

the relation between learning methods and academic achievement. This was the aim of the present experiment.

We hypothesized that WM could either mediate or moderate the effects of Jigsaw on performance. On the one hand, WM may act as a mediator (Hypothesis 1). Learning new concepts can be highly demanding for all students, whatever their individual WM capacity. If Jigsaw classroom does alleviate the individual cognitive load during the collective learning phase, then greater individual WM resources should be available to master the main content, resulting in higher individual performance. We thus directly assessed WM to estimate its mediating role between Jigsaw method and performance. We also measured student subjective mental workload, using the NASA-TLX questionnaire, to compare with prior studies and results obtained with the WM test.

Alternatively, WM could act as a moderator (Hypothesis 2). Without directly releasing executive resources, the positive interdependence of Jigsaw classroom might help counterbalance lower WM. Each student has a defined role for collective learning, actively listens to others, can find help from experts for knowledge structuring, and gains from repeated exposure to pedagogical content in written and oral formats. This arrangement could enhance learning for low WM students. Additionally, consistent with findings showing that the expert phase drives Jigsaw's positive effects [44,45], we examined whether individual WM moderates this influence.

We tested our hypotheses in a naturalistic field study among undergraduates enrolled in an introductory course on critical thinking and logical fallacies, an ideal context for our investigation. As a novel subject at the university, critical thinking can pose cognitive load challenges as students try to understand and differentiate between various fallacies. In line with Deiglmayr and colleagues [46], Jigsaw classroom was compared to a cooperative learning method with goal, but no resource interdependence (i.e., weak-knowledge interdependence condition). In this control condition, the goal of the cooperation was similar to the Jigsaw condition (i.e., "the objective is to produce a collective work with your group, by creating new examples of fallacies"). However, there was no resource interdependence (i.e., no division of the pedagogical material between students). This common university practice is less structured than Jigsaw, lacking the same degree of interdependence, as all students receive complete pedagogical content and are working in groups without additional guidelines.

## Method

### Participants

We used a large convenience sample of first-year undergraduate science students (N = 342; 12 classes ranging from 20 to 33 students). This sample was particularly appropriate for assessing the effects of the Jigsaw classroom in higher education settings, where the method is commonly used. Students voluntarily participated in the study as part of a new general university course, "Methodology", which focused on learning strategies. The 12 classes were randomly assigned to either the Jigsaw condition or the cooperative control condition ($n$ = 164 and $n$ = 178 *students*, respectively). Each class was divided into small groups of 4-5 students, resulting in 80 groups (38 Jigsaw groups, 42 control groups). Due to exclusion criterion on the WM task, 64 participants were removed from the analyses, leading to a final sample of 278 participants (171 female students, $M$age =19.28, $SD$ =.957). As we used a convenience sample, classes were recruited based on their availability, not *a priori* sample-size calculation. A post-hoc sensitivity analysis simulation indicated that the smallest difference between group slopes our sample could detect was an effect size ($|\Delta$ slope$|$) of 0.33 (two-tailed α =.05, power =.817), and revealed an empirical power of 0.645, 95% CI = [0.636, 0.655].

## Materials

The learning materials (i.e., handouts) were developed by the third author, an expert in teaching critical thinking. They were tailored to align with the content division required by the Jigsaw method. Handouts (i.e., Descartes, Hypatia, Schopenhauer, and Socrates) contained definitions of sophisms and two logical fallacy examples (e.g., the slippery slope). Overall, eight categories of logical fallacies were either split into four handouts (i.e., two logical fallacies in the Jigsaw condition) or gathered in one handout (eight logical fallacies in the control condition).

## Measures

### Working memory complex-span task

We used the Automated Symmetry Span task (ASSPAN, [47]) to measure individual WM. Participants first completed three practice sessions (storage task, processing task, and both interleaved tasks). Then they were serially presented with a series of red squares within a 4x4 matrix and had to remember their positions (the storage task). Set sizes ranged from 2 to 5 items to recall per trial in a 4x4 blank matrix, across twelve trials (i.e., three trials for each set size). Each square presentation was followed by a symmetry judgement task (the processing task), where participants decided whether a pattern presented in an 8x8 matrix was symmetrical on its vertical axis. Tasks were shown sequentially on the screen, with participants responding by tapping the tablet. In the storage task, participants recalled the location and order of red squares. In the processing task, they had to keep their symmetry rate equal to or better than 85%. The WM task was programed with Inquisit 5 Web (Millisecond Software, 2018) and completed on iPad 5 tablets (Apple) with iOS 11.

Following previous research [48], participants failing below 80% on the processing component (i.e., the symmetry task), either on the WM baseline and/or the WM test, were excluded from data analysis. This ensured attention allocation on both storage and processing components of the task. This criterion excluded 18,71% (N = 64) participants from analyses. Three participants with missing WM baseline data, but who completed the rest of the experiment were retained for analysis. Absolute span score was used as the outcome, summing perfectly recalled sets with correct order. Given the three trials for each set size of 2, 3, 4, and 5 items, the absolute span score varied from 0 to 42.

### Perception of the cognitive load

We used the NASA-RTLX scale [49] to measure students' subjective mental workload. This multidimensional instrument is particularly adapted to represent the combination of workload one might experience during a task [50] and presents excellent psychometric properties [51]. Participants rated mental load during collective production on six subscales: effort, frustration, mental demand, physical demand, temporal demand, and performance. Responses used a 21-point scale from 1 (very low) to 21 (very high). Following established practice and the recommendations of Byers and colleagues [49], we analyzed each subscale individually and did not conduct an internal consistency analysis, as this would misrepresent the scale's design.

### Individual critical thinking performance

Performance was assessed with a 10 multiple-choice quiz adapted from the "Ennis-Weir critical thinking essay test" [52]. It evaluated individual understanding of eight logical fallacies. For instance, "Nuclear weapons are a nuisance as the Nobel Prize winner Georges Charpak,

who has worked for years to promote nuclear energy, has stated loud and clear" had to be identified as an 'argument from authority'. Each correct response was scored 1. The total score was computed multiplying correct answers by 2 and varied from 0 to 20. For one class, output for two questions was not recorded due to a technical issue, so their total score was multiplied by 10/8 for rescaling. No prior knowledge of critical fallacies was measured, as the content of the course was part of a newly implemented course at the university (i.e., Methodology), meaning that students were allegedly never exposed to this content. Furthermore, the random assignment of participants to the two experimental conditions should have neutralized any pre-existing differences in critical thinking, especially given the large sample size. It should also be noted that the content of the lesson was identical in both conditions.

## Collective critical thinking performance

A collective production task, inspired by the Ennis-Weir critical thinking essay test [52], was employed as an incentive to work in small groups in both experimental (Jigsaw method) and control (voluntary cooperation without resource interdependence) conditions. Each of the 80 small groups created four fallacies from randomly assigned categories of logical fallacies (e.g., slippery slope) to defend a quirky statement (e.g., 'retirees are responsible for global warming'). Co-authors, blind to conditions, scored responses. Success was coded as 1, failure or missing argument as 0, resulting in a percentage score ranging from 0 to 100. As for the individual critical thinking quiz, no prior students' performance on this task was available.

## Procedure

All students, regardless of condition (Jigsaw or control), were informed of working in small groups (4-6 students) on logical fallacies and being individually tested on their understanding at the end of the class.

In the *Jigsaw condition*, both resource and goal interdependence were implemented. First, jigsaw groups were formed in alphabetical order to prevent friendship bias, and each member of the group read their handout individually (5 minutes). In groups composed of 5 members, two Socrates handouts were distributed (there was no group of six students in the Jigsaw condition). Second, students met in the expert groups (15 to 20 minutes), sharing information and ideas about new examples of their assigned logical fallacies. Then, they returned to their jigsaw groups, teaching (as experts) their assigned fallacies and learning (as novices) the other fallacy categories from their peers (30 minutes). By the end, each student had grasped all eight logical fallacies. In the *voluntary cooperative (control) condition*, goal but no resource interdependence was implemented. The following instructions were given: "*During this class, you will work into small groups and the functioning of your memory will be evaluated. You will develop your critical mind by learning about some reasoning biases, that we call "fallacious arguments". You will discover them and learn how to debunk them. The final objective is to produce a collective work with your group, by creating new examples of fallacies. After this group activity, we will ask you again to take the Working Memory test, the same you took last time, and to individually answer some questions to check you properly understood the class.*" Students received the entire pedagogical content, formed small groups (using the alphabetical order), read individually the handout presenting the eight fallacy categories (20 minutes), then discussed the logical fallacies in-group (40 minutes). The complete instructions are detailed in the online Supporting Information.

Testing both WM's mediating and moderating roles required two measurements. The first (baseline) measurement occurred one to three weeks before the experimental session to test for the moderation hypothesis (the moderator must not be impacted by the experimental

manipulation) and ensure comparable WM capacity across conditions. This baseline session lasted from 20-30 minutes. Students received instructions to create an alphanumerical ID and were provided with tablets to complete the WM task. The second (test) measurement occurred in the experimental session to test the mediation hypothesis ($M_{time}$ between testing = 18.5 days). During this session, students worked in small groups (Jigsaw or voluntary cooperation) to grasp logical fallacy categories, completed the WM test, generated new fallacy examples as a group (collective performance), answered the cognitive load questionnaire, and took the individual quiz.

This 2-hour experimental session was conducted during a class with teachers following the Jigsaw or control instructions. Two experimenters aided teachers in material distribution, WM task instructions, and ensuring compliance with conditions. Tablets were used for the WM test, cognitive load questionnaire, and quiz. After the group activities, participants of both conditions individually completed the WM task on tablets ($M$ = 8.58 minutes, $SD$ = 2.32). All groups then had 20 minutes to complete the collective production of new fallacy examples. Finally, students completed the cognitive load questionnaire and the multiple-choice quiz on fallacies. In both conditions, teachers answered students' questions but gave little feedback. See sections S1 and S2 of the online Supporting Information for details on the instructions and the scoring procedure.

## Analytic Strategy

We assessed WM's mediation and moderation in Jigsaw's impact on critical thinking using two linear mixed-effects models (LMM, [53]) with groups set as random intercept since individual observations were nested into small groups (ICC=.196, 95% CI = [.067,.316]). Learning Condition (Jigsaw vs. control) was set as the independent variable, and performance on the Critical Thinking quiz was the outcome in both models. Experimental WM score was intended as mediator, and prior (baseline) WM score as moderator. Our mediation analysis was based on the index approach to mediation proposed by Hayes [54], that focuses on testing the indirect path directly, without requiring that all individual paths be significant. For moderation analysis, we followed Hayes' [54] approach, which argues that a significant overall relationship between the independent and dependent variables is not necessary to test for moderation.

Another model was performed to differentiate Jigsaw's effects on subtopic performance during the expert and novice phases, considering individual WM capacity. In the Jigsaw classroom's final phase, students acted as both experts (teachers) and novices (learners). A generalized linear mixed-effects model with binomial error structures (GLMM) was performed, using response accuracy (probability of correct answers on the quiz's ten items) as the dependent variable. Fixed-effect predictors were WM score, Subtopic (4 levels: Descartes, Hypatia, Schopenhauer, Socrates), and their interactions with Learning condition. Subjects, groups, and items were set as random intercepts. The full model was compared with a null model, including random predictors only and WMC baseline as covariate. Contrasts were made between the three learning conditions (experts vs. novices vs. control).

Statistical analyses were run in R 4.0.1 (R core team 2020). Model parameters were bootstrapped using 10000 repetitions, and confidence intervals were computed using the adjusted bootstrap percentile method at 95% [2.5%, 97.5%]. Models were fitted using the function glmer of the package *lme4* ([53]). Reported β values are standardized coefficients (centered mean, one standard deviation unit). To probe the interaction, simple slopes comparisons were performed with Bonferroni's adjustment method using the *emmeans* and *emtrends* packages. In all the comparisons, "estimate" stands for the difference between estimated marginal means. Statistical significance was set at α =.05.

Finally, considering the challenges discussed in the literature regarding standardized effect sizes in linear mixed models – particularly the complications introduced by variance partitioning [55] – we report effect sizes using standardized beta coefficients. This approach, common in mixed model analyses, ensures statistical rigor and reproducibility while appropriately addressing the nested data structure and random effects [56].

## Open practices

All measures, manipulations, and data/participant exclusions are reported in the manuscript or the online Supporting Information. Data supporting the findings of this study are openly available at the Open Science Framework repository: https://osf.io/2vcqf/?view_only=1cfd8b5d09a045e7ae8fda852c6bb960.

While not pre-registered, we present our original hypotheses and results, including those that yielded no significant findings, to ensure their reproducibility and facilitate future replication studies.

## Results

Table 1 presents dependent variable correlations and reliability coefficients (Cronbach's α), and Table 2 reports samples descriptive statistics and mean difference effect sizes (Cohen's $d$). WM task test-retest reliability was moderate ($r = .42$, $p < .01$). Calculated at the level of individual item [57], the internal consistency of the WM absolute spans was high (α =.78: baseline session; α = 77: experimental session).

### Working memory as a mediator of the Jigsaw classroom effects

We hypothesized that Jigsaw classroom's structured information sharing would free up individual WM resources, leading to improved performance (Hypothesis 1). This hypothesis would be supported by a positive indirect path of Learning Condition on the critical thinking performance through WM measured during class (experimental session). Our mediation model was tested while controlling for WM baseline. Results showed that Learning Condition was not a significant predictor of students' performance (see Table 3). More importantly, our mediation hypothesis was rejected due to the absence of indirect (estimate = -0.004, 95% CI [-0.025, 0.00]) and total effect (estimate = 0.082, 95% CI [-0.040, 0.191]) of Learning Condition on performance. Only a significant effect of WM was observed on critical thinking performance.

**Table 1. Descriptive statistics and Pearson correlations for study variables.**

| Measure | M | SD | Range | Skew | Kurtosis | α | 1 | 2 | 3 | 4 | 5 | 6 |
|---|---|---|---|---|---|---|---|---|---|---|---|---|
| 1. Critical Thinking Quiz | 11.95 | 4.94 | 0 – 20 | -.260 | -.668 | .70 | – | | | | | |
| 2. Collective Production | 45.14 | 28.56 | 0 – 100 | .153 | -.896 | – | .06 | – | | | | |
| 3. Absolute Span Baseline | 21.43 | 8.71 | 0 – 42 | .008 | -.631 | .78 | .11 | -.08 | – | | | |
| 4. Absolute Span Test | 25.26 | 8.70 | 4 – 42 | -.234 | -.682 | .77 | .15* | .05 | .42** | – | | |
| 5. Symmetry Rate Baseline | 92.64 | 4.87 | 81 – 100 | -.518 | -.395 | – | .07 | -.02 | .35** | .20** | – | |
| 6. Symmetry Rate Test | 93.29 | 5.15 | 81 – 100 | -.669 | -.319 | – | .14* | -,00 | .08 | .30** | .33** | – |

N=278.

*p <.05. **p <.01. As variables 2, 5, and 6 were not scale variables but were expressed in percentages, Cronbach's α were not computed.

**Table 2. Means, Standard Deviations, and t-test statistics for Jigsaw and Control conditions.**

| Variable | Jigsaw | | Control | | t-value | Cohen's *d* |
|---|---|---|---|---|---|---|
| | **M** | **SD** | **M** | **SD** | | |
| Working Memory Capacity | | | | | | |
| Absolute Span - Baseline | 21.19 | 8.91 | 21.72 | 8.50 | -.499 | .060 |
| Absolute Span -Test | 24.81 | 8.66 | 25.79 | 8.75 | -.935 | .113 |
| Cognitive load (NASA-RTLX) | | | | | | |
| Mental demand | 12.89 | 4.02 | 12.60 | 3.96 | .615 | -.074 |
| Physical demand | 3.03 | 2.82 | 3.37 | 3.54 | -.882 | .106 |
| Temporal demand | 12.13 | 4.86 | 9.91 | 5.02 | 3.722 *** | -.448 |
| Performance | 14.03 | 4.27 | 15.40 | 4.08 | -2.711** | .327 |
| Effort | 12.52 | 3.93 | 12.01 | 3.74 | 1.109 | -.134 |
| Frustration | 7.82 | 5.67 | 8.02 | 5.73 | -.295 | .036 |
| Critical thinking performance | | | | | | |
| Quiz | 12.27 | 4.21 | 11.57 | 5.67 | 1.17 | -.141 |
| *Collective Task* – Rate (0-100) | 44.86 | 28.18 | 45.4724 | 29.11 | -.176 | .021 |

N = 278 (Jigsaw = 151, Control = 127).

**p <.01. ***p <.001.

**Table 3. Results from the mediation model predicting critical thinking quiz.**

| Effect | | Estimated Individual performance | CI [2.5%, 97,5%] | SE |
|---|---|---|---|---|
| **direct effect** | | | | |
| $\beta_a$ | LC -> WMC | -.044 | [-.155,.062] | .054 |
| $\beta_{c'}$ | LC -> quiz | .086 | [-.035,.190] | .058 |
| **$\beta_b$** | **WMC -> quiz** | **.077 *** | **[.008,.204]** | **.051** |
| | $WMC_{Baseline}$ -> quiz | .040 | [-.067,.154] | .058 |
| Indirect effect | | | | |
| | LC -> WMC -> quiz | -.003 | [-.025,.000] | .004 |
| Total effect | | | | |
| $\beta_{c'}+\beta_{ab}$ | | .082 | [-.043,.186] | .058 |

CI= confidence interval. For effect in bold, 95% CI excludes zero. LC: Learning Context. WMC: Working Memory Capacity.

*p <.05. **p <.01.

## Working memory as a moderator of the Jigsaw classroom effects

We assumed that Jigsaw classroom could help low WM students to better perform the pedagogical content (Hypothesis 2). To ensure equality between Jigsaw and control group, a LMM analysis was first performed with Subjects set as random effect, Learning Condition (Jigsaw vs. Control) and Session (baseline and experimental WM sessions) as predictors, and WM as the outcome. No intergroup differences were found, neither at baseline (estimate = 0.059, 95% CI [-.173,.290], *p* =.618) nor experimental session (estimate = 0.110, 95% CI [-.122,.341], *p* =.351), suggesting similar WMC in both conditions. However, results revealed a significant WM improvement from session 1 to 2 (estimate = - 0.431, 95% CI [-0.556, -0.307], *p* <.001).

Due to this unanticipated test-retest effect, our analytical strategy required adjustment. Scharfen and colleagues [58] outlined that test-retest effects on complex WM tasks can be due to several "debilitating construct-irrelevant factors" (e.g., test anxiety, lack of familiarity with

the task, administration in real-life setting), and recommend using retest scores instead of baseline scores to provide a more accurate WM capacity estimate and avoid misinterpretation. As our retest WM scores were not affected by the learning condition (see section above), we followed this recommendation. Despite this, for transparency, we first present results using the original strategy (WM baseline score), followed by those using retest scores.

Analyses using WM baseline as moderator revealed no significant effect on the individual critical thinking performance (see S1 Fig in the online Supporting Information, $\chi^2(1) = 0.420$, $p =.516$). WM baseline score (estimate = 0.120, SE = 0.087, $p =.168$), Learning Condition (estimate = 0.167, SE = 0.145, $p =.252$), and their interaction (estimate = - 0.074, SE = 0.11, $p =.525$) did not reach significance.

Analyses using retest WM score as moderator and controlling for WM baseline score (consistent results were obtained with and without this covariate) revealed a significant interaction effect between Learning Condition and WM (see Fig 1) on individual critical thinking performance, $\chi^2(1) = 5.716$, $p =.017$ (see Table 4). Simple slopes analyses showed that lower WM participants performed better in Jigsaw than in control condition (simple slope at WM -1SD: 0.448, 95% CI = [0.090, 0.806], SE =.183 $p =.016$), whereas higher WM students performed similarly in both conditions (simple slope at WM +1SD: -0.099, CI = [-0.455, 0.257], SE =.182, $p =.587$). The Jigsaw advantage for low WM students was supported by the other simple slopes: Higher critical thinking performance depended on higher WM in control condition (simple slope of.236, CI = [0.059, 0.412], SE =.090, $p =.009$), but not in Jigsaw condition (simple slope of -0.037, CI = [-0.200, 0.125], SE =.082, $p =.649$).

The results of a supplementary moderation analysis with the processing component of the WM task are also reported in S1 Table and S3 Fig in the online Supporting Information.

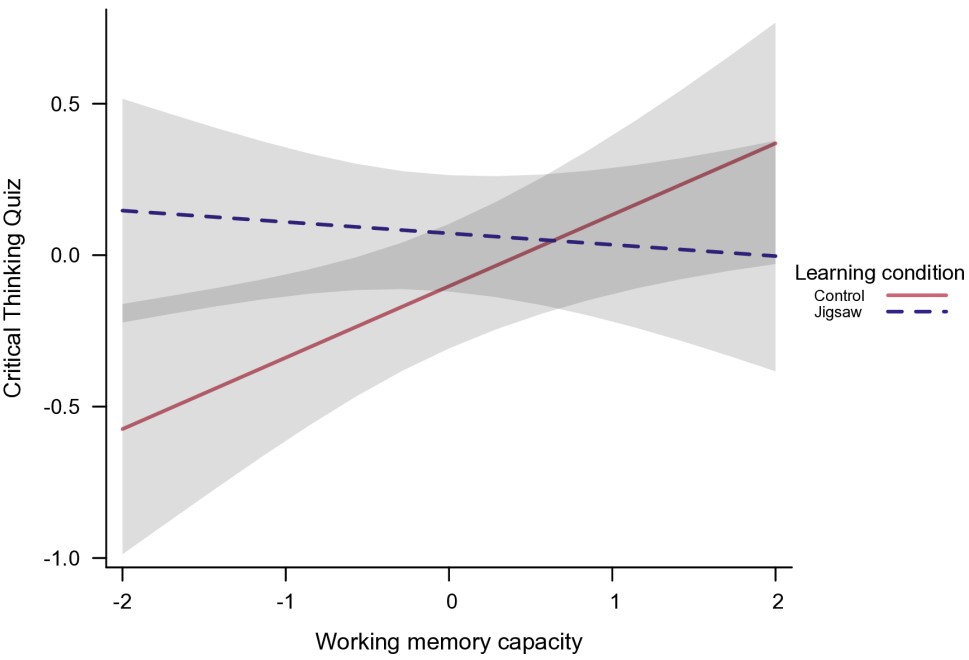

**Fig 1. Interaction effect between Learning Condition and Working Memory capacity on individual quiz performance.** All values are standardized (z-scores), error bands indicate 95% confidence intervals.

**Table 4. Results from the moderation model predicting critical thinking quiz.**

| Effect | Estimated Individual performance | CI [2.5%, 97,5%] | SE |
|---|---|---|---|
| Fixed effects | | | |
| Intercept | -.102 | [-.306,.101] | .103 |
| $\beta_{WMC}$ | **.235 \*\*** | **[.062,.410]** | **.089** |
| $\beta_{LC}$ | .174 | [-.102,.456] | .141 |
| $\beta_{Interaction}$ | **-.273 \*** | **[-.497, -.051]** | **.115** |
| $\beta_{Baseline}$ | .043 | [-.084,.173] | .063 |
| Random effects | | | |
| Within-groups variance | .157 | [.040,.309] | |
| Between-groups variance | .806 | [.649,.960] | |

CI= confidence interval. For effects in bold, 95% CIs exclude zero. LC: Learning context. WMC: Working Memory Capacity.

\*p <.05. \*\*p <.01.

### Expert and novice phases of the Jigsaw method

A GLMM analysis was performed to disentangle Jigsaw's effects on expertise and novice phases in subtopic performances based on WM. Three groups were compared: expert and novice groups from Jigsaw and students from the control condition. The comparison between the full and the null model indicated that the combined predictors, Subtopic, WM, and their interaction with Learning Context had a significant effect on items accuracy, $\chi^2(14) = 59.63$, $p <.001$. Pairwise contrasts revealed that experts outperformed novices (estimate = 0.699, 95% CI = [0.339, 1.061], $p <.001$) and control students (estimate = 0.774, 95% CI = [0.249, 1.298], $p =.001$). Consistent with the main analyses, higher WM correlated with higher accuracy scores only among control students, $\beta = 0.348$, 95% CI = [0.123, 0.574], $p =.003$ (see Fig 2). Moreover, no difference was observed between control and novice students regarding high WM, estimate = -0.307, 95% CI = [-0.892, 0.279], $p =.629$.

### Perceived cognitive load

Finally, a LMM assessed subjective cognitive load variations across learning conditions. Results indicated significant differences between Jigsaw and Control conditions on two Nasa-RTLX dimensions: Jigsaw students reported higher temporal demand (estimate = 0.414, SE=0.149, 95% CI = [0.203, 0.662], $p =.007$) and lower perceived performance (estimate = -0.315, SE = 0.134, 95% CI = [-0.532, -0.108]), $p =.021$).

### Discussion

The present study investigated working memory (WM) as a cognitive mechanism underlying the effects of Jigsaw classroom on students' critical thinking achievement. Our findings showed that WM acted as a moderator (not a mediator), and thereby provided useful information to disentangle previous inconsistent results [12], in specifying for whom the Jigsaw classroom is likely to benefit. Most studies on the relationship between working memory, learning and academic achievement [26,27] generally indicate the limitations of students with low WMC. Our findings illustrate that a specific cooperative learning setting like the Jigsaw classroom can mitigate the academic performance gap for low-WM students.

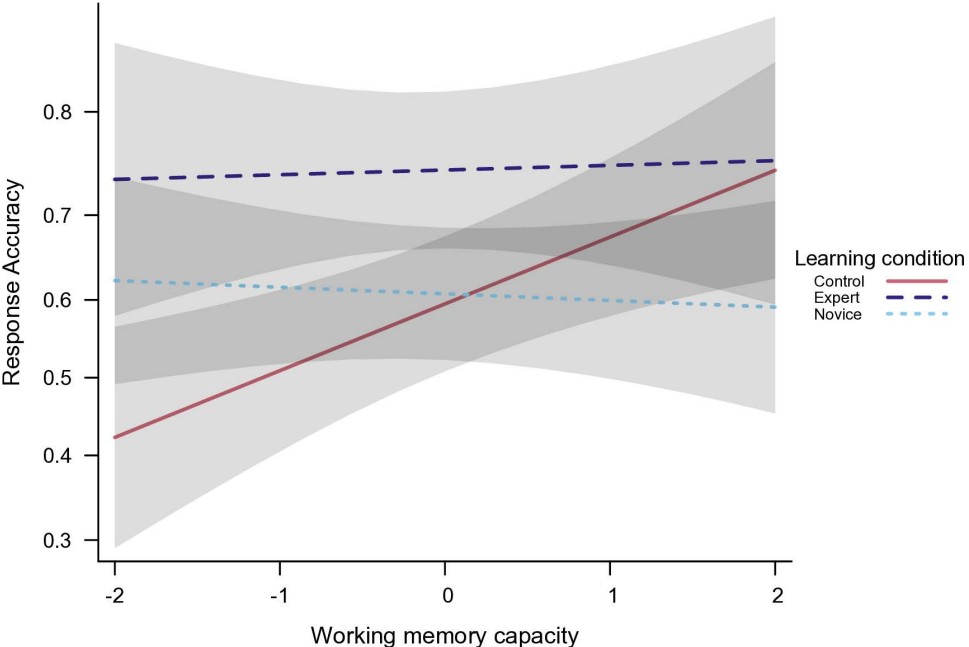

**Fig 2. Interaction effect between Learning context (Control vs. Experts vs. Novices) and Working Memory capacity on individual quiz performance.** All values are standardized (z-scores), error bands indicate 95% confidence intervals. Y-axis is truncated for greater visibility.

## The mediating role of WM in Jigsaw effects

Our expectation of improved individual performance through higher WM scores in the Jigsaw condition after the group activity was not corroborated, refuting our mediation hypothesis. This could be attributed to the critical thinking task's limited cognitive load stimulation, possibly concealing cognitive load differences in control students (voluntary cooperation). Another explanation could be that any reduction in task cognitive load due to information sharing in the Jigsaw Classroom was countered by transaction costs —increased cognitive efforts dedicated to interindividual communication and coordination. Janssen et al. (2010) proposed such costs could counter cooperation benefits, imposing extraneous cognitive load. Our subjective cognitive load assessment (NASA-TLX) findings seem consistent with this suggestion: Interestingly, in contrast to Kirschner et al. [22] but consistent with Nebel et al. [16], Jigsaw students reported amplified temporal demands during collective learning, despite equal pedagogical activity time (2 hours). This heightened extraneous load, linked to unfamiliar Jigsaw structure, aligned with lower success expectations reported by students compared to the control condition (see Table 2).

Regarding other potential mediators of Jigsaw effects, Nebel et al. [16] found that collective performance mediated the effect of their intervention (interdependence vs. voluntary cooperation) on individual performance. They suggested that the social interdependence of Jigsaw favored more interactions and group performance, leading to enhanced individual learning outcomes. This was not the case in our study, as illustrated by the weak correlation between group and individual performance ($r =.06$, $p =.282$) and the lack of an indirect effect of group performance (supplementary analyses, $\beta= -0.002$, 95% CI [- 0.063, 0.026]). Although our data do not establish the causal role of WM in Jigsaw's effects, more research is necessary to unravel the cognitive processes underlying cooperative learning, especially on long-term

memory issues. It is also reasonable to consider social predictors (e.g., self-esteem or self-efficacy) for capturing the variance of the Jigsaw effects on academic achievement.

## The moderating role of WM in Jigsaw effects

Our results suggest that the Jigsaw classroom benefited low WM students, without impacting high WM students' performance. Through Jigsaw, low WM students not only outperformed their peers in the unstructured cooperative learning condition but also performed as well as high WM students. This effect represents a two-point increase in the average score, a substantial improvement within the context of the French education system's 0-20 scoring metric. Moreover, these findings are in line with previous research demonstrating Jigsaw's positive impact only on low-skilled students [18] or students from minorities [59].

Given Jigsaw's complex nature (resource interdependence, task specialization, peer-teaching), isolating a single factor to explain its positive impact on low WM students is challenging. We suggest that fundamental memory processes might be at play. Jigsaw's social learning environment (peer-teaching) demanded active engagement with information, and the sequential learning phases (individual, expert, jigsaw) facilitated repeated exposure to pedagogical content. These repetitions likely bolstered information retrieval for students who typically struggle with encoding essential information. The critical thinking task used here involved unfamiliar pedagogical material that can be considered as moderately difficult (mean score of 12/20). Our findings align with Janssen et al. [20], suggesting collaborative learning's benefits for demanding tasks and potential drawbacks for low-demand tasks or proficient students. However, Retnowati and colleagues (experiment 1, [24]) found that in high-complexity tasks (like transfer tasks), collaborative learning could yield lower performance than individual learning. Further research could examine whether Jigsaw's positive effects for low WM students depend on task difficulty.

## The role of the expert phase in the Jigsaw classroom

We conducted additional analyses to compare experts, novices, and control students. Results showed a substantial higher performance for experts within their assigned materials, reinforcing the pivotal role of this phase in driving Jigsaw effects [44–45]. This expert phase likely induced a double coding of the information (episodic and semantic), improving retention for students with special educational needs and potentially facilitating knowledge construction by the cognitive system.

## Limitations

Some limitations of this study should be considered. Notably, administering the WM task in a real-life classroom context, rather than a tightly controlled lab setting, introduces certain considerations. This might have influenced: i) the comparatively lower baseline WM score in contrast to the subsequent measurement, ii) the moderate test-retest correlation ($r = .43$), and iii) the limited correlation between the initial baseline WM measurement and academic performance. The 4-point difference between our two WM (baseline and test) measurements suggests an important practice effect. Following Scharfen and colleagues' [58] recommendations, we deemed our WM (test) measure was a more accurate reflection of students' capacities. Paradoxically, this limitation underscores the importance of conducting research in real-life settings. Our study is a notable instance of WM assessment within a classroom context, providing a meaningful contribution compared to highly controlled lab environments that may have exaggerated the ecological relevance of their findings (for a similar argumentation, see Friso-van den Bos and colleagues [60]).

Concerning WM's moderating role, one might contend that Jigsaw classroom's benefits for low WM students stemmed from their potential for improvement relative to high achievers. However, this explanation contradicts our findings. Higher WM students' mean performances did not exceed 13 on a 0 to 20 scale, indicating room for improvement for them as well (see S2 Fig in the online Supporting Information). Prior research has also demonstrated the challenge of achieving academic performance gains through educational intervention, particularly among low achievers [10,61].

Moreover, instructional methods can yield varying effects for low and high achievers [62,63]. For example, whereas high instructional guidance (characteristic of Jigsaw classroom) is necessary for low achievers, it can be counterproductive for higher achievers who thrive with fewer constraints to leverage their learning strategies and knowledge base [64]. Consequently, our findings bridge a gap in understanding the prerequisites for Jigsaw classroom's effectiveness, highlighting its potential to assist struggling students when tailored support is provided.

## Implications and future directions

Our study of WM's role in the Jigsaw classroom sheds light on the interaction between human cognitive architecture and group learning environments [19], offering a promising approach for instructing low achievers. The Jigsaw classroom seems to benefit students with WM limitations, who are often at risk for educational underachievement [65]. Further research could explore whether this interdependent cooperative learning, particularly the expert phase, could similarly benefit students with diverse learning needs or disabilities (e.g., ADHD or dyslexia).

Jigsaw classroom presents an alternative to prevalent cognitive trainings often prescribed for these groups to address their challenges (for a critical review see Melby-Lervåg and colleagues [66]), especially in remote learning contexts. Rather than solely focusing on temporary WM enhancement and expecting widespread academic improvements, we can reasonably anticipate positive advancements by introducing structure and interdependence within the learning environment, whether in physical or virtual classrooms.

## Conclusion

As one of the first experimental examinations of WM's role in the interplay between Jigsaw classroom and academic achievement, our study showed how cognitive capacity differences can modulate Jigsaw's impact on academic outcomes. Our findings highlighted that low WM students achieved better critical thinking learning task following a single Jigsaw classroom session. Furthermore, we elucidated how Jigsaw's peer-teaching aspect during the expert phase served as a pivotal factor in this beneficial effect, supporting and broadening prior findings. At the theoretical level, our results hold significance for cognitive, social, and educational psychology, unveiling insights into the black box of the Jigsaw Classroom. They also offer practical implications for educators, informing them about whom and why the Jigsaw method can improve academic achievement.

## Supporting information

**S1. Supp_WM_Jigsaw.docx.**
(DOCX)

## Acknowledgements

We thank all the teachers and undergraduates who participated in this study. We would also like to thank the collaborators from the PROFAN consortium for their valuable insights and support in this study. The PROFAN consortium is composed of: Batruch Anatolia, Bouet

Marinette, Bressan Marco, Bressoux Pascal, Brown Genavee, Butera Fabrizio, Cepeda Carlos, Cherbonnier Anthony, Darnon Céline, Demolliens Marie, De Place Anne-Laure, Desrichard Olivier, Ducros Théo, Goron Luc, Hemon Brivael, Huguet Pascal, Jamet Eric, Martinez Ruben, Mazenod Vincent, Mella-Barraco Nathalie, Michinov Estelle, Michinov Nicolas, Ofosu Nana, Pansu Pascal, Peter Laurine, Petitcollot Benoit, Poletti Céline, Régner Isabelle, Riant Mathilde, Robert Anaïs, Rudmann Ocyna, Sanrey Camille, Stanczak Arnaud, Toumani Farouk, Vilmin Simon, Visintin Emilio Paolo and Vives Eva.

**Note:** all consortium members were provided access to the manuscript before its submission to Plos One and have been informed about the forthcoming submission. All members are in agreement with the ICMJE guidelines.

## Author contributions

**Conceptualization:** Eva Vives, Céline Poletti, Isabelle Régner.

**Data curation:** Marco Bressan.

**Formal analysis:** Eva Vives, Marco Bressan.

**Investigation:** Eva Vives, Céline Poletti, Denis Caroti, Isabelle Régner.

**Methodology:** Eva Vives, Céline Poletti, Isabelle Régner.

**Resources:** Denis Caroti, Isabelle Régner.

**Writing – original draft:** Eva Vives, Marco Bressan.

**Writing – review & editing:** Fabrizio Butera, Pascal Huguet, Isabelle Régner.

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
