## [Decision Letter · Decision Letter 0]

4 Oct 2024

PONE-D-24-28685Uncovering the relationship between working memory and performance in the cooperative Jigsaw classroomPLOS ONE

Dear Dr. Vives,

Thank you for submitting your manuscript to PLOS ONE. After careful consideration, we feel that it has merit but does not fully meet PLOS ONE’s publication criteria as it currently stands. Therefore, we invite you to submit a revised version of the manuscript that addresses the points raised during the review process.

We look forward to receiving your revised manuscript.

Kind regards,

Sheikh Arslan Sehgal, PhD

Academic Editor

PLOS ONE

Journal Requirements:

4. We note that the grant information you provided in the ‘Funding Information’ and ‘Financial Disclosure’ sections do not match. When you resubmit, please ensure that you provide the correct grant numbers for the awards you received for your study in the ‘Funding Information’ section.

5.  Thank you for stating the following financial disclosure: “This work was supported by the French Ministry of National Education, Youth and Sports (MENJS); the Ministry of Higher Education, Research and Innovation (MESRI); the « Mission Monteil pour le numérique éducatif », and the « Programme d’investissements d’avenir, expérimentation ProFAN» (PIA).”

6. Please note that your Data Availability Statement is currently missing the repository name. If your manuscript is accepted for publication, you will be asked to provide these details on a very short timeline. We therefore suggest that you provide this information now, though we will not hold up the peer review process if you are unable.

7. When completing the data availability statement of the submission form, you indicated that you will make your data available on acceptance. We strongly recommend all authors decide on a data sharing plan before acceptance, as the process can be lengthy and hold up publication timelines. Please note that, though access restrictions are acceptable now, your entire data will need to be made freely accessible if your manuscript is accepted for publication. This policy applies to all data except where public deposition would breach compliance with the protocol approved by your research ethics board. If you are unable to adhere to our open data policy, please kindly revise your statement to explain your reasoning and we will seek the editor's input on an exemption. Please be assured that, once you have provided your new statement, the assessment of your exemption will not hold up the peer review process.

8. Please ensure that you include a title page within your main document. You should list all authors and all affiliations as per our author instructions and clearly indicate the corresponding author.

9. One of the noted authors is a group or consortium “PROFAN consortium”. In addition to naming the author group, please list the individual authors and affiliations within this group in the acknowledgments section of your manuscript. Please also indicate clearly a lead author for this group along with a contact email address.

10. Please include captions for your Supporting Information files at the end of your manuscript, and update any in-text citations to match accordingly. Please see our Supporting Information guidelines for more information: http://journals.plos.org/plosone/s/supporting-information. 

Reviewers' comments:

Reviewer's Responses to Questions

**Comments to the Author**

1. Is the manuscript technically sound, and do the data support the conclusions?

Reviewer #1: No

Reviewer #2: Yes

2. Has the statistical analysis been performed appropriately and rigorously? 

Reviewer #1: No

Reviewer #2: Yes

3. Have the authors made all data underlying the findings in their manuscript fully available?

Reviewer #1: Yes

Reviewer #2: Yes

4. Is the manuscript presented in an intelligible fashion and written in standard English?

Reviewer #1: Yes

Reviewer #2: Yes

5. Review Comments to the Author

Reviewer #1: Thank you for the opportunity to review the manuscript entitled: "Uncovering the relationship between working memory and performance in the cooperative Jigsaw classroom”.

Currently, in the education more than teaching processes and the teacher’s work in the classroom, it is the learning processes through which students achieve their proposed objectives in each subject that capture most interest. In accordance with this new approach centered on learning, the use of methodologies based on active constructive learning such as cooperative learning. This methodology enables students to acquire basic skills and increases their motivation to participate actively in the learning process. These days, learning methods based on cooperation are more and more widely used with the aim of encouraging teamwork, allowing students to learn to work as part of a team, improving performance and learning and developing interpersonal skills.

The paper examined whether working memory capacity mediated or moderated the effects of Jigsaw classroom on individual performance. Also, the findings offer insight into the potential cognitive mechanisms implied in the success of the Jigsaw method and provide new recommendations for educators on how to redeem the deficit of low working-memory capacity students on performance.

Introduction:

Cooperative learning has been widely investigated since the 1970s, when the first studies on specific implementations emerged. According to Gillies (2016), several metha-analyses have provided ample evidence of the effectiveness of cooperative learning on students’ outcomes: learning and performance. Some investigations have focused mainly on issues related to the efficacy of cooperative learning and the mediating mechanisms involved. The goal of these investigations revolves around two axes. The first one is the nature and quality of the interactive process. The second axis refers to prior factors that condition the efficacy of cooperative learning.

The authors suggest that taking into account potential moderating and mediating cognitive variables would help to understand Jigsaw mixed findings by specifying for whom and how the Jigsaw method can benefit (or not) to academic achievement. Given the Jigsaw method's reliance on resource interdependence, investigating working memory (WM) becomes pertinent.

Why can cognitive variables such as working memory be mediators or moderators? There are many variables such as learning goals, self-concept, anxiety, expectations, attitudes towards teamwork, social skills, cognitive style…, that can be mediators or moderators. The introduction does not provide sufficient background. It lacks a robust conceptual framework.

Participants:

The participants were not chosen at random. The number of participants from a university is not representative and directly affects external validity.

Instruments:

The instruments must always display two important qualities: reliability and validity. The authors should have calculated the reliability coefficients for their sample (NASA-RTLX scale) and should have calculated McDonald's Omega, Composite Reliability (CR) and Average Variance Extracted (AVE) of the subscales (effort, frustration, mental demand, physical demand, temporal demand, and task success). Cronbach's Alpha is conditioned by the number of items and the number of alternative responses, so it is necessary to use other alternative reliability indices, such as McDonald's Omega which is calculated through factorial loads and are measured more accurate reliability.

Did the “task success-perfomance” scale use range from 1 from 1 (very low) to 21 (very high)?

Results:

Effect sizes differences should have been calculated using Cohen´s d or Hedge´s g. (Table 2)

Mediation Model Analysis (Table 3)

The mediation model must meet the requirements to conduct a simple mediation analysis: significant relations between the independent (LC) and the dependent (Quiz) variables (c= total effect), between the independent (LC) variable and the mediator (VMC, a), and between the mediator (VMC) and the dependent variable (Quiz, b). Additionally, the b score is larger than c´ (direct effect), and c` is smaller than c.

In this study there are no significant relations (a: LC-VMC, β=- .044, p>0.05; c: LC-Quiz, β=- .082, p>0.05)

If the conditions were not met, why did the authors perform the mediation analysis? It makes no sense.

Moderation Model Analysis (Table 4)

In this study there are no significant relationships between the independent (LC) and dependent (Quiz) variable (β=- .174, [-.102, .456]). It makes no sense to look for a moderator variable between two variables that are not significantly related.

We do not know conditional effects of the focal predictor at values of the moderator.

References:

Lack of up-to-date references. The authors only present three current references (last five years 2020-2024).

Thank you.

Reviewer #2: It is an interesting and thought provoking research. I wonder of whether the relationship between condition (Jigsaw or control) and critical thinking performance is moderated specifically by WMC. Both developmental cognitive psychology (e.g., Fry & Hale, 2000) and cognitive aging literature (e.g., Verhaeghen, 2011 -metaanalysis of over 100 studies; Salthouse, 1996) indicate that the processing speed is related directly and by mediation of working memory to complex cognition. The ASSPAN measure of working memory consist of the two tasks: The storage task and the processing task. As I understand the storage task was used as the principal measure of WMC in the moderating analysis (Table 4). I wonder of whether the same results would be obtained when Symmetry Rate Test instead of Absolute Span Test (this symmetry judgment task seems to be closely related to the processing speed task) was applied as moderating variable? Interestingly (see Table 1) the correlations between Critical Thinking Quiz and both Absolute Span Test and Symmetry Rate Test were significant and almost identical.

Therefore I'd suggest that Authors discuss the possibility that the Jigsaw Classroom could enhance performance not only students with low working memory capabilities but also students with slow processing speed.

6. PLOS authors have the option to publish the peer review history of their article (what does this mean? ). If published, this will include your full peer review and any attached files.

**Do you want your identity to be public for this peer review?** For information about this choice, including consent withdrawal, please see our Privacy Policy .

Reviewer #1: **Yes: ** Benito León del Barco

Reviewer #2: No

---

## [Author Response · Author response to Decision Letter 1]

22 Nov 2024

1. Is the manuscript technically sound, and do the data support the conclusions?

Reviewer #1: No

Reviewer #2: Yes

2. Has the statistical analysis been performed appropriately and rigorously?

Reviewer #1: No

Reviewer #2: Yes

3. Have the authors made all data underlying the findings in their manuscript fully available?

Reviewer #1: Yes

Reviewer #2: Yes

4. Is the manuscript presented in an intelligible fashion and written in standard English?

Reviewer #1: Yes

Reviewer #2: Yes

5. Review Comments to the Author

Author’s response: We would like to thank both the Reviewers for their comments as well as their suggestions for improving our manuscript. We would also like to thank the Editor for their interest in our manuscript. You will find a point-by-point response to the issues raised by the Reviewers. Responses are written in blue font in our letter and in the revised version of the manuscript.

Reviewer #1: Thank you for the opportunity to review the manuscript entitled: "Uncovering the relationship between working memory and performance in the cooperative Jigsaw classroom”.

Currently, in the education more than teaching processes and the teacher’s work in the classroom, it is the learning processes through which students achieve their proposed objectives in each subject that capture most interest. In accordance with this new approach centered on learning, the use of methodologies based on active constructive learning such as cooperative learning. This methodology enables students to acquire basic skills and increases their motivation to participate actively in the learning process. These days, learning methods based on cooperation are more and more widely used with the aim of encouraging teamwork, allowing students to learn to work as part of a team, improving performance and learning and developing interpersonal skills.

The paper examined whether working memory capacity mediated or moderated the effects of Jigsaw classroom on individual performance. Also, the findings offer insight into the potential cognitive mechanisms implied in the success of the Jigsaw method and provide new recommendations for educators on how to redeem the deficit of low working-memory capacity students on performance.

R1: Introduction: Cooperative learning has been widely investigated since the 1970s, when the first studies on specific implementations emerged. According to Gillies (2016), several metha-analyses have provided ample evidence of the effectiveness of cooperative learning on students’ outcomes: learning and performance. Some investigations have focused mainly on issues related to the efficacy of cooperative learning and the mediating mechanisms involved. The goal of these investigations revolves around two axes. The first one is the nature and quality of the interactive process. The second axis refers to prior factors that condition the efficacy of cooperative learning.

The authors suggest that taking into account potential moderating and mediating cognitive variables would help to understand Jigsaw mixed findings by specifying for whom and how the Jigsaw method can benefit (or not) to academic achievement. Given the Jigsaw method's reliance on resource interdependence, investigating working memory (WM) becomes pertinent.

Why can cognitive variables such as working memory be mediators or moderators? There are many variables such as learning goals, self-concept, anxiety, expectations, attitudes towards teamwork, social skills, cognitive style…, that can be mediators or moderators. The introduction does not provide sufficient background. It lacks a robust conceptual framework.

Author's response: We agree with the reviewer that the above variables may be relevant mediators and moderators of cooperative learning on achievement. Indeed, a recent review (Authors et al., 2024) found that learning goals, self-concept, and attitudes towards Jigsaw classroom have received considerable interest from practitioners and academics. However, little or no work to date, has explored the relationships between individuals' cognitive abilities and cooperative learning in the classroom. Given that working memory capacity is a well-established predictor of academic performance and learning capacity during a course, it was critical to explore this "third variable" in the context of the Jigsaw classroom. The Jigsaw method, which uniquely relies on resource interdependence and structured peer teaching, demands considerable cognitive engagement from students. As detailed in the section Inside the Black Box of Jigsaw Classroom, we argue that examining WM may thus enable us to address a notable gap in cooperative learning research by illuminating how cognitive resources interact with this specific instructional approach. Our focus on working memory aligns with current research on cognitive load in collaborative learning (Zambrano et al., 2019; Kirschner et al., 2018), contributing to a deeper understanding of how cooperative methods may differentially benefit learners based on cognitive capacity.

We have, hopefully, made our explanation clearer in the manuscript on page 6.

R1: Participants: The participants were not chosen at random. The number of participants from a university is not representative and directly affects external validity.

Author’s response: We acknowledge the reviewer’s point regarding the non-random nature of our sample and the limitations it may pose for generalizing findings beyond the university context. As with many studies conducted among university students, our sample may not be representative of the general population.

However, our research was designed as a field study to examine the effects of the Jigsaw method among undergraduate students within a natural classroom setting, rather than in a controlled laboratory environment. A random sample of participants from the general population would not have met our research objectives, which were to test the effectiveness of this cooperative learning approach in a university context where it is commonly applied.

Moreover, we tested a very large sample of students (n = 342 participants). This sample size falls within the higher range of samples commonly used in this area of research. On average, studies conducted with Jigsaw classroom have used relatively smaller sample sizes. Data from a recent systematic review (Authors et al., 2024) showed that of the 69 studies included in the review, the average sample size was 119.37 participants. This means that our sample size is 187% larger than the average reported in the review and is positioned in the 97th percentile of sample sizes. Our sample is thus notably larger, enhancing the robustness and reliability of our findings within the university context.

Finally, we made sure that different disciplines were represented by having classes in biology, statistics and economics. A total of 12 classes were enrolled in the study, from different sites of the faculty. This diversity contributes to the heterogeneity of our sample and supports the ecological validity of our findings within higher education settings.

For clarity and transparency purposes, we have inserted an explanation about our sample on page 8 of the manuscript: “We used a large convenience sample of first-year undergraduate science students (N = 342; 12 classes ranging from 20 to 33 students). This sample was particularly appropriate for testing the effects of the Jigsaw classroom in higher education settings, where the method is commonly used. Students voluntarily participated in the study as part of a new general university course, “Methodology”, which focused on learning strategies”.

R1: Instruments: The instruments must always display two important qualities: reliability and validity. The authors should have calculated the reliability coefficients for their sample (NASA-RTLX scale) and should have calculated McDonald's Omega, Composite Reliability (CR) and Average Variance Extracted (AVE) of the subscales (effort, frustration, mental demand, physical demand, temporal demand, and task success). Cronbach's Alpha is conditioned by the number of items and the number of alternative responses, so it is necessary to use other alternative reliability indices, such as McDonald's Omega which is calculated through factorial loads and are measured more accurate reliability.

Author’s response: We thank the Reviewer for their comments and their request to provide the psychometric properties for the NASA-RTLX scale. We agree with the Reviewer when they say Cronbach’s alpha would not be an appropriate reliability parameter. Indeed, assessing internal consistency of this questionnaire does not align with the scale design, as the NASA-RTLX measures six different types of workloads, not a unidimensional construct. Calculating the McDonald’s Omega, for similar reasons, would not provide useful information about reliability, as the six dimensions of the NASA-RTLX are not conceptually expected to correlate as highly as items on a unidimensional scale.

A more appropriate measure of reliability for the NASA-RTLX would be test-retest reliability or an intra-class correlation coefficient (ICC) to assess the stability of responses across time for each dimension independently, as reported in the literature (e.g. Devos et al., 2020; Rubio et al., 2004). However, such an analysis was beyond the scope of the present study. In line with existing studies using the NASA-RTLX, we followed established practice by focusing on each dimension individually without conducting an internal consistency analysis, as this would misrepresent the scale's design.

This information is now presented on page 10: “Following established practice and the recommendations of Byers et al. (1989), we analyzed each subscale individually and did not conduct an internal consistency analysis, as this would misrepresent the scale’s design”. The reliability parameters of the working memory measures were also reported in Table 1, on page 32. This is now indicated clearly on page 14: “Table 1 presents dependent variable correlations and reliability coefficients (Cronbach’s α), and Table 2 reports samples descriptive statistics and mean difference effect sizes (Cohen’s d)”.

R1: Did the “task success-perfomance” scale use range from 1 from 1 (very low) to 21 (very high)?

Author’s response : Yes, as with the five other NASA-TLX subscales, the performance subscale used a 21-point scale (from 1-very low to 21-very high). This item specifically asked participants, “Do you believe you have successfully completed the tasks required for this assignment?” (in French in the study). This approach aligns with the standard use of NASA-TLX scales, ensuring consistency across workload dimensions.

For clarity we have changed the name of this subscale to “performance” (see on page 10) which is congruent with the original publication (Byers et al., 1989).

R1: Results

R1: Effect sizes differences should have been calculated using Cohen´s d or Hedge´s g. (Table 2)

Author’s response: We thank the Reviewer for this valuable suggestion and have now reported Cohen’s d to quantify the differences between the Jigsaw and control groups. This change is now indicated on page 15 and appears in Table 2, on page 33.

Additionally, we have included an explanation of our approach to reporting effect sizes in mixed models on page 14 “Finally, considering the challenges discussed in the literature regarding standardized effect sizes in linear mixed models—particularly the complications introduced by variance partitioning (Pek & Flora, 2018)—we report effect sizes using standardized beta coefficients. This approach, common in mixed model analyses, ensures statistical rigor and reproducibility while appropriately addressing the nested data structure and random effects (Lorah, 2018). Differences between the Jigsaw and control groups are also provided in Table 2 in the form of Cohen’s d.

References

Lorah, J. (2018). Effect size measures for multilevel models: Definition, interpretation, and TIMSS example. Large-scale assessments in education, 6(1), 1-11.

Pek, J., & Flora, D. B. (2018). Reporting effect sizes in original psychological research: A discussion and tutorial. Psychological methods, 23(2), 208.

R1: Mediation Model Analysis (Table 3). The mediation model must meet the requirements to conduct a simple mediation analysis: significant relations between the independent (LC) and the dependent (Quiz) variables (c= total effect), between the independent (LC) variable and the mediator (VMC, a), and between the mediator (VMC) and the dependent variable (Quiz, b). Additionally, the b score is larger than c´ (direct effect), and c` is smaller than c.

In this study there are no significant relations (a: LC-VMC, β=- .044, p>0.05; c: LC-Quiz, β=- .082, p>0.05). If the conditions were not met, why did the authors perform the mediation analysis? It makes no sense.

Author’s response: We appreciate the reviewer's feedback and understand the concern about our approach to mediation analysis. We recognise that the Reviewer's comments are based on the classical mediation framework proposed by Baron and Kenny (1986), which requires certain conditions to be met before conducting a mediation analysis. However, our analysis is based on a more contemporary theoretical approach, namely the index approach to mediation, proposed by Hayes (2013, 2017, 2022). Unlike the causal steps approach of Baron and Kenny, Hayes’ approach focuses on testing the indirect effect directly, without requiring that all individual paths be significant. This perspective, which has been widely adopted, is considered both statistically robust and flexible. It allows for mediation to be established through an indirect pathway that can exist independently of a significant total effect (Hayes & Rockwood, 2020). This is particularly relevant in cases where complex mechanisms may still operate through an indirect pathway, which aligns with our theoretical framework for exploring underlying cognitive mechanisms in the Jigsaw method. This modern approach recognises that meaningful mediation can occur in the absence of significant direct relationships, thereby extending the applicability of mediation analysis to a wider range of data structures. We hope this clarifies our methodological choice and its alignment with current statistical practice.

Furthermore, in support of transparency and reproducibility in science, we chose to fully report our argumentation process - from the development of our theoretical hypothesis of mediation to the detailed presentation of our statistical results. We believe that sharing our mediation analysis, even in cases where results were not statistically significant, is essential in the context of the current reproducibility crisis (Franco et al., 2014). Our aim w

---

## [Decision Letter · Decision Letter 1]

14 Jan 2025

PONE-D-24-28685R1Uncovering the relationship between working memory and performance in the Jigsaw classroomPLOS ONE

Dear Dr. Vives,

Thank you for submitting your manuscript to PLOS ONE. After careful consideration, we feel that it has merit but does not fully meet PLOS ONE’s publication criteria as it currently stands. Therefore, we invite you to submit a revised version of the manuscript that addresses the points raised during the review process.

We look forward to receiving your revised manuscript.

Kind regards,

Sheikh Arslan Sehgal, PhD

Academic Editor

PLOS ONE

Journal Requirements:

Reviewers' comments:

Reviewer's Responses to Questions

**Comments to the Author**

1. If the authors have adequately addressed your comments raised in a previous round of review and you feel that this manuscript is now acceptable for publication, you may indicate that here to bypass the “Comments to the Author” section, enter your conflict of interest statement in the “Confidential to Editor” section, and submit your "Accept" recommendation.

Reviewer #2: All comments have been addressed

Reviewer #3: (No Response)

2. Is the manuscript technically sound, and do the data support the conclusions?

Reviewer #2: Yes

Reviewer #3: Partly

3. Has the statistical analysis been performed appropriately and rigorously? 

Reviewer #2: Yes

Reviewer #3: Yes

4. Have the authors made all data underlying the findings in their manuscript fully available?

Reviewer #2: Yes

Reviewer #3: Yes

5. Is the manuscript presented in an intelligible fashion and written in standard English?

Reviewer #2: Yes

Reviewer #3: Yes

6. Review Comments to the Author

Reviewer #2: Thank you for addressing my concerns. In my opinion this paper is interesting and thought provoking.

Reviewer #3: I have read the paper and the comments from a previous round of reviews. I enjoyed reading the paper which I think is well written, and I think the authors have done a good job of addressing the reviewer comments. I only have one comment that I think does need to be addressed before publication.

I think the authors need to exercise more caution in their conclusion that WM mediates the effect of the Jigsaw classroom. I think this for two reasons.

1. I appreciate that the authors have justified their use of the index approach to mediation. But I think there are still reasonable arguments for why you would not look for a mediation effect without a direct effect. I think it is fine to use the index approach, given there clearly are papers to back up this method. But I do think it might necessitate a bit more caution in their interpretation.

2. Much more importantly, the authors run the mediation analysis twice. Once with the original pre-registered measure of WM (pre-experiment) and one using the measure taken during the experiment which was intended to be used for the moderation analysis. I appreciate that the authors have been entirely transparent in this, and have explained and justified why they have done this, and I welcome that. However, their conclusion then states that WM is found to mediate the effect of the Jigsaw classroom. And what they have in fact found is that one analysis does show a mediation effect, and one doesn't. Again, I appreciate they suggest that the second measure of WM is more reflective of the 'real' WMC due to possible anxiety re: the original test and unfamiliarity with the test. Whilst I am sympathetic to a degree with those arguments, the test they used is designed to be administered in one sitting as a valid measure of WM. And so, I think there needs to be more acknowledgement that, overall, their results suggest there *may* be a mediating role for WM, but that it is not conclusive. And I think that needs changing in the abstract and in the Discussion.

7. PLOS authors have the option to publish the peer review history of their article (what does this mean? ). If published, this will include your full peer review and any attached files.

**Do you want your identity to be public for this peer review?** For information about this choice, including consent withdrawal, please see our Privacy Policy .

Reviewer #2: No

Reviewer #3: No

---

## [Author Response · Author response to Decision Letter 2]

15 Jan 2025

6. Review Comments to the Author

Reviewer #2: Thank you for addressing my concerns. In my opinion this paper is interesting and thought provoking.

We would like to thank you again for taking the time to read our manuscript and the comments we addressed.

Reviewer #3: I have read the paper and the comments from a previous round of reviews. I enjoyed reading the paper which I think is well written, and I think the authors have done a good job of addressing the reviewer comments. I only have one comment that I think does need to be addressed before publication.

I think the authors need to exercise more caution in their conclusion that WM mediates the effect of the Jigsaw classroom. I think this for two reasons.

1. I appreciate that the authors have justified their use of the index approach to mediation. But I think there are still reasonable arguments for why you would not look for a mediation effect without a direct effect. I think it is fine to use the index approach, given there clearly are papers to back up this method. But I do think it might necessitate a bit more caution in their interpretation.

2. Much more importantly, the authors run the mediation analysis twice. Once with the original pre-registered measure of WM (pre-experiment) and one using the measure taken during the experiment which was intended to be used for the moderation analysis. I appreciate that the authors have been entirely transparent in this, and have explained and justified why they have done this, and I welcome that. However, their conclusion then states that WM is found to mediate the effect of the Jigsaw classroom. And what they have in fact found is that one analysis does show a mediation effect, and one doesn't. Again, I appreciate they suggest that the second measure of WM is more reflective of the 'real' WMC due to possible anxiety re: the original test and unfamiliarity with the test. Whilst I am sympathetic to a degree with those arguments, the test they used is designed to be administered in one sitting as a valid measure of WM. And so, I think there needs to be more acknowledgement that, overall, their results suggest there *may* be a mediating role for WM, but that it is not conclusive. And I think that needs changing in the abstract and in the Discussion.

We would like to thank you for the review of our manuscript and your comments. We would like to address one particular point in order to ensure the clarity of our manuscript. With regard to your statement 'I think the authors need to exercise more caution in their conclusion that WM mediates the effect of the Jigsaw classroom' and your first remark, it is important to clarify that we did not observe any mediation effect in our study. Although the mediation hypothesis was tested, the model resulted in an absence of indirect and total effect on the student's performance. This is indicated on page 11 in the Results section and in the Discussion section on page 19. However, we did observe that the effects of Jigsaw cooperative learning were moderated by the working memory capacity, leading to better performance among students with lower WMC. This effect was observed on the retest WM score, set as moderator (see pages 17-18 and Figure 1). For this reason, we concluded that our moderation hypothesis was supported by the data. We agree that nevertheless, these findings, as any other work, must call further replication and confirmation. We therefore nuanced our statement and rephrased our claim in the Discussion section. You can see the changes made on pages 20 and 23: ‘Our results suggest that the Jigsaw classroom benefited low WM students, without impacting high WM students’ performance’ and ‘The Jigsaw classroom seems to benefit students with WM limitations, who are often at risk for educational underachievement (e.g., Dunning et al., 2013)’. And on the Abstract on page 3: ‘Multilevel analyses revealed that working memory capacity moderated—but did not mediate—the effect of the Jigsaw classroom. That is, Jigsaw enhanced performance for students with low working memory capacities’.

---

## [Decision Letter · Decision Letter 2]

4 Feb 2025

Uncovering the relationship between working memory and performance in the Jigsaw classroom

PONE-D-24-28685R2

Dear Dr. Vives,

We’re pleased to inform you that your manuscript has been judged scientifically suitable for publication and will be formally accepted for publication once it meets all outstanding technical requirements.

Kind regards,

Sheikh Arslan Sehgal, PhD

Academic Editor

PLOS ONE

Additional Editor Comments (optional):

Reviewers' comments:

Reviewer's Responses to Questions

**Comments to the Author**

1. If the authors have adequately addressed your comments raised in a previous round of review and you feel that this manuscript is now acceptable for publication, you may indicate that here to bypass the “Comments to the Author” section, enter your conflict of interest statement in the “Confidential to Editor” section, and submit your "Accept" recommendation.

Reviewer #3: All comments have been addressed

2. Is the manuscript technically sound, and do the data support the conclusions?

Reviewer #3: Yes

3. Has the statistical analysis been performed appropriately and rigorously? 

Reviewer #3: Yes

4. Have the authors made all data underlying the findings in their manuscript fully available?

Reviewer #3: Yes

5. Is the manuscript presented in an intelligible fashion and written in standard English?

Reviewer #3: Yes

6. Review Comments to the Author

Reviewer #3: First I would like to apologise for the confusion in my review. I realise I had become confused with the mediation and moderation, and that was not helpful.

However, my point that you have run one of the analyses twice, and once found an effect and once not, does still stand. As does my point about your justification for why you believe the second WM test is more reflective of their WM ability. I am sympathetic to the arguments you make about this. However, the test you used is not designed to *need* a re-test. And so we are still in the situation where you measure WM and using that WM data you do not find a moderation effect. Then you measure WM again and using that data you do find a moderation effect. Which suggests to me that this may not be a very robust finding. Of course all research needs replication. But in this paper you have run the moderation analysis twice, and you have not replicated your findings within this paper. Which, for me, suggests the need for more caution than the usual caveats relating to the fact that all research needs replication. So my own view is that I think this requires a little more caution than you have in this second review. However, I appreciate that you have added in some qualifying statements around the finding, so thank you for doing that. And I do think this research is interesting, important, and well written. So I will leave it up to the Editor if they would like to encourage a little more caution. But am also happy to recommend it is accepted as it is.

7. PLOS authors have the option to publish the peer review history of their article (what does this mean? ). If published, this will include your full peer review and any attached files.

**Do you want your identity to be public for this peer review?** For information about this choice, including consent withdrawal, please see our Privacy Policy .

Reviewer #3: No

---

## [Editor Report · Acceptance letter]

PONE-D-24-28685R2

PLOS ONE

Dear Dr. Vives,

I'm pleased to inform you that your manuscript has been deemed suitable for publication in PLOS ONE. Congratulations! Your manuscript is now being handed over to our production team.

Kind regards,

on behalf of

Dr Sheikh Arslan Sehgal

Academic Editor

PLOS ONE